# ACEtimation—The Combined Effect of Adverse Childhood Experiences on Violence, Health-Harming Behaviors, and Mental Ill-Health: Findings across England and Wales

**DOI:** 10.3390/ijerph20176633

**Published:** 2023-08-23

**Authors:** Rebekah Lydia Miriam Amos, Katie Cresswell, Karen Hughes, Mark A. Bellis

**Affiliations:** 1School of Medical and Health Sciences, Bangor University, Wrexham LL13 7YP, UK; katie.cresswell@bangor.ac.uk (K.C.); m.a.bellis@ljmu.ac.uk (M.A.B.); 2World Health Organization Collaborating Centre on Investment for Health and Well-Being, Public Health Wales, Wrexham LL13 7YP, UK; karen.hughes18@wales.nhs.uk; 3Faculty of Health, Liverpool John Moores University, Liverpool L2 2ER, UK

**Keywords:** adverse childhood experiences, childhood adversity, child maltreatment, household dysfunction, mental health, health-harming behaviors, violence, cross sectional, survey data

## Abstract

Adverse childhood experiences (ACEs) encompass various adversities, e.g., physical and/or emotional abuse. Understanding the effects of different ACE types on various health outcomes can guide targeted prevention and intervention. We estimated the association between three categories of ACEs in isolation and when they co-occurred. Specifically, the relationship between child maltreatment, witnessing violence, and household dysfunction and the risk of being involved in violence, engaging in health-harming behaviors, and experiencing mental ill-health. Data were from eight cross-sectional surveys conducted in England and Wales between 2012 and 2022. The sample included 21,716 adults aged 18–69 years; 56.6% were female. Exposure to child maltreatment and household dysfunction in isolation were strong predictors of variant outcomes, whereas witnessing violence was not. However, additive models showed that witnessing violence amplified the measured risk beyond expected levels for being a victim or perpetrator of violence. The multiplicative effect of all three ACE categories demonstrated the highest level of risk (RRs from 1.7 to 7.4). Given the increased risk associated with co-occurring ACEs, it is crucial to target individuals exposed to any ACE category to prevent their exposure to additional harm. Implementing universal interventions that safeguard children from physical, emotional, and sexual violence is likely to mitigate a range of subsequent issues, including future involvement in violence.

## 1. Introduction

Adverse childhood experiences (ACEs) are potentially traumatic events affecting children. They may include experiences such as child maltreatment (e.g., physical, sexual, and emotional abuse), witnessing domestic violence, parental substance use, and exposure to war [1]. Early exposure to ACEs can have profound negative impacts on an individual’s physical health, mental health, and mortality risk. ACEs have been linked to an increased risk of engaging in health-harming behaviors such as alcohol use, sexual risk-taking and interpersonal violence [2]. They have been found to be strongly associated with mental health conditions such as depression and anxiety [2,3] and have also been linked to weight-related disorders such as dysmorphic disorder [4] and obesity [4]. Exposure to ACEs can subsequently impact physical health throughout the life course, including through early development of non-communicable diseases such as cancer and cardiovascular disease [2]. Accordingly, evidence suggests that there is greater use of health services by adults with ACEs [2,5,6]. Evidence also indicates the intergenerational transmission of ACEs, along with maternal mental health issues post-partum [3]. Thus, ACEs not only affect the individuals who experience them but can potentially impact their own children.

Understanding the future risks associated with different forms of ACE exposure facilitates the development of tailored and effective intervention strategies [7,8]. To this end, ACEs have been useful in public health research to understand the antecedents of disease and map the predictive likelihood of risky health behaviors and future disease in the general population [2,8]. It is unlikely that all ACEs contribute to poor outcomes equally; moreover, this effect is unlikely to remain consistent across different types of outcomes, for example, sexual risk-taking versus mental ill-health [9]. Previous research has identified considerable heterogeneity in the strength of the relationship between a high ACE count and various health-related issues and behaviors in adulthood [2]. As such, the type of ACE experienced may have differing effects on health outcomes, with certain ACE combinations attenuating or amplifying these effects.

Combinations of exposure to ACE categories may have additively detrimental associations. For example, when multiple ACE categories are experienced within the same individual, they may confer excess risk over and above the effects of any one ACE category in isolation [10]. Researchers have attempted to estimate this excess risk utilizing additive models; for example, analysis of a national cohort in the US revealed that sexual abuse amplified the negative effect seen on outcomes when combined with other ACEs, being most detrimental to mental ill-health in adulthood [11]. A similar study utilizing a large US national sample explored again how ACEs could amplify the negative effects of each ACE. Here, experiences of sexual abuse and physical abuse were associated most strongly with behavioral problems in young people [12].

However, in ACE research, cumulative scores are often used as indicators of increasing risk, where the number of ACEs someone experiences is summed. A score of four or more ACEs is typically considered “high risk” [7]. This can be an issue for two reasons; firstly it assumes a linear and cumulative effect, i.e., that ACEs, irrespective of type, have a quantitatively similar (growing) effect on the individual. Secondly, the summing method does not allow the estimation of how much each ACE type confers risk in isolation and/or when experienced in combination. ACE exposures are often interrelated, with certain types of ACE more associated than others [13]. Researchers have categorized ACEs into subtypes to allow nuanced analyses whilst maintaining a pragmatic approach to data analysis and interpretation. For example, ACEs have been categorized into child maltreatment and household dysfunction, where the former includes more direct forms of harm or deprivation, and the latter includes disruptive environments and, thus, is an indirect form of ACE [14]. A recent study exploring the differential effects of ACE categorizations, namely child maltreatment and household dysfunction, found that the former was more predictive of depression and anxiety [14]. This highlights that different ACE categories may confer differing levels of risk. Thus, there is potential benefit in analyzing ACEs by category.

Arguably, experiencing violence directly through child maltreatment has a different qualitative experience on an individual than witnessing violence towards someone else (e.g., parents) [15,16]. Furthermore, household dysfunction, which is more concerned with environmental unrest and parental functioning, is again qualitatively different from witnessing violence or being exposed to child maltreatment. There is likely to be a crossover in these experiences, and previous studies have found subtypes of child maltreatment to be highly correlated with good convergent validity across measures [17]. Thus, ACEs may be organized into the following categories: child maltreatment, witnessing violence, and household dysfunction.

Currently, there are no estimates of the effect of child maltreatment, witnessing violence, and household dysfunction in isolation and combination with health behaviors and outcomes in English and Welsh populations. The current study utilized a large general population sample from England and Wales to explore the association between ACE categories and a range of outcomes, including health-related behaviors, violence, and mental ill-health. The primary aim of the study was to estimate the associated effect of each ACE category in isolation and in combination.

Our primary hypothesis was that exposure to any individual ACE category or multiple ACE categories would be significantly associated with experiences of violence, health-harming behaviors, and mental ill-health. We expected the strength of this relationship to vary depending on the combination of ACE categories experienced and the type of outcome being measured. We had no specific predictions as to which ACE categories would be the most/least associated with any of the measured outcomes. The secondary hypothesis was that there would be an additive effect when ACE categories co-occurred; specifically, when ACE categories were experienced simultaneously, the measured effects would be over and above those expected statistically.

## 2. Materials and Methods

### 2.1. Data and Sample Characteristics

Data were collected via eight cross-sectional studies conducted across various locations in Wales and England between 2012 and 2022, utilizing face-to-face, online, and telephone interview methods. Survey stratification and recruitment approaches for each study are detailed in Appendix A. Professional market research companies were commissioned to conduct household and telephone surveys, with interviewers operating to the Market Research Society Code of Conduct. Participants were included if they were adults and were cognitively able to participate.

For most household surveys (England 2012, England 2013, England 2015, Wales 2017), individual households were identified via the national postcode address file. Selected households were sent a study opt-out letter along with study information. This differed slightly for the Wales 2015 study, where interviewers instead randomly selected households and provided study information at the door. For telephone surveys (England and Wales 2022), telephone contacts (landline and mobile) were obtained from a commercial sample provider and, in Wales, by stratified Welsh Health Board area. Recruitment for online surveys used a commercial online panel consisting of individuals who engage in online research for compensation. No incentives were provided for face-to-face or telephone interviews. Before participating, all participants provided informed consent and were informed that they were free to withdraw at any time. Questions of a sensitive nature were completed by the participant themselves, and participant responses were anonymized.

The analyzed sample consisted of individuals who provided complete demographic and ACE data. The analytical sample was also limited to those aged 18–69 years, as certain studies focused specifically on this age range (see Appendix A). There was a proportionally equal split of the total sample in each age bracket (lowest 18.2% [50–59 years of age]—highest 19.6% [18–29 years of age]). The sex distribution in the sample was also balanced, with women accounting for 56.6% of respondents. The final sample size consisted of 21,716 participants.

### 2.2. Measures

Demographic data included measures of ethnicity, age, sex, and indices of multiple deprivation (IMDs). Participants were asked which ethnic group they belonged to (White, Asian, or Asian British, Black/African/Caribbean/Black British, mixed, or other). Given the low representation of ethnicities in categories other than White, the ethnicity variable used here was collapsed into two categories—White or other than White. IMDs are standardized scores of deprivation across different areas and regions; this standardized score was converted into deprivation quintiles and provided an overall indicator of deprivation for Welsh [18] and English areas [19].

ACEs were measured via the Centers for Disease Control and Prevention short ACE tool [20]. Participants were asked if they experienced nine different ACEs before 18 years of age: physical abuse, sexual abuse, emotional abuse, divorce, witnessing domestic violence, household exposure to alcohol abuse, drug abuse, mental ill-health, and incarceration. To estimate ACE combinations efficiently, we grouped ACEs into three distinct categories: witnessing violence (domestic violence), exposure to child maltreatment (physical, sexual, and emotional abuse), and household dysfunction (substance misuse, alcohol misuse, mental illness, and incarceration). Participants were included in the child maltreatment or household dysfunction categories if they had experienced at least one of the ACEs within that category (e.g., for child maltreatment, physical, sexual, and/or emotional abuse). To note, divorce as an ACE was not included in any of the ACE categories. This is because it accounts for a large proportion of the sample experiencing ACEs and would likely skew the results. Furthermore, it was not considered appropriate to include divorce in the household dysfunction category, as divorce does not necessarily denote dysfunction. As such, eight ACE variables were used to derive the ACE categories used in the analyses.

Outcome variables included being a perpetrator of violence (within the last 12 months), being a victim of violence (within the last 12 months), being incarcerated (ever), engagement in binge drinking (e.g., consumption of 5/6 drinks or more), cannabis use (ever), current smoking, having had a sexually transmitted infection (STI) (ever), early sexual initiation (<16 years), accidental teenage pregnancy (<18 years of age), low life satisfaction (current), and any mental illness diagnosis (ever). The full wording of all questions used and the categorization of outcome variables are specified in Appendix A.

There was variation between surveys in the outcome questions asked (see Appendix A) and, thus, the analyzable sample size varied across outcomes. For instance, in the England 2013 and England (South) 2015 surveys, 10 out of the 11 outcome variables were available. On the other hand, the Wales 2022 survey only had responses for 2 out of the 11 outcomes (See Appendix A).

### 2.3. Analytical Strategy

The predictor variable had eight levels representing all ACE category combinations (e.g., No ACEs (reference category): witnessed violence only, witnessed violence and household dysfunction, all three ACE categories). This variable was entered into a series of generalized linear models—with a binomial/Poisson distribution and log function—as a dichotomous predictor with the 11 outcome variables (victim of violence, perpetrator of violence, incarcerated, smoking, binge drinking, cannabis use, STI, early sexual initiation, accidental teenage pregnancy, low life satisfaction, and any mental illness diagnosis). All analyses controlled for the effects of study (i.e., survey location and stratification), sex, age, ethnicity, and deprivation quintile (see Appendix A for analyses by demographic variables).

Multiplicative interaction terms have commonly been used to estimate the joint effect of two risk factors [9]; however, this is usually done on a relative odds scale (logistic regression), and any absence of effect does not mean the absence of any clinically relevant interaction [21]. Thus, additive interactions have also been used to identify whether the joint effects of two risk factors are significantly greater than the sum of the individual risk factors allowing estimation of the excess risk observed over and above the combined individual risk of each exposure [21,22]. Modern epidemiologists posit that estimating additive interaction effects can be more clinically informative as it suggests that the effect of one exposure is dependent on the presence of the other. We utilized the methods outlined by Andersson et al. (2005) to calculate these additive effects using generalized linear models. The first step was to develop disjoint categories which reflected all possible ACE combinations. As Andersson’s method is designed to handle a maximum of three combinations, they make three categories of the predictor variable (01, 10, 11) and a reference category (0). We explored eight possible ACE combinations and, thus, created eight levels of this predictor (001, 011, 010, 110, 100, 111, 101, reference category 000). Whereas Andersson used regression coefficients to calculate additive risk, we instead used estimated marginal means obtained from generalized linear models (IBM, knowledge center). This was to identify the increased likelihood of risk to those exposed to differing levels of the exposure (for more information regarding methods of calculation, please see Appendix A). It is also entirely possible that when ACE exposures are combined, they are not additive at all and, in some cases, may be associated with lesser risk (antagonistic). Hypothetically exposure to any one ACE category (e.g., child maltreatment) could be so potent that any further ACE exposure has a diminished effect. Given the focus of our hypotheses, we did not explore such effects as to do so effectively would be much more informative if we had temporal data regarding ACE exposures (i.e., the order of when they occurred). The SPSS (version 27, IBM Statistics, New York, NY, United States) code will be made available upon request.

## 3. Results

The demographic characteristics of the sample are detailed in Table 1. Sample sizes across ACE category exposures are presented in Figure 1. Overall, 12,952 (59.6%) had 0 ACEs and 8764 (40.4%) had been exposed to at least one ACE category.

### 3.1. Multiplicative Models

As can be seen in Table 2a,b, the associated effect of each single ACE category varied across outcomes in terms of strength and significance. For example, child maltreatment and household dysfunction were associated with all outcomes, whereas the witnessing violence category was associated with the least number of outcomes (3/11 outcomes); being a victim of violence, being incarcerated, and ever using cannabis.

The strongest relationship between any single ACE category exposure and outcome was exposure to child maltreatment and being a perpetrator of violence as an adult. Those experiencing child maltreatment were almost three times more likely to report being a perpetrator of violence as an adult.

Compared to other ACE categories, child maltreatment was most strongly associated with accidental teenage pregnancy (RR = 1.8), low life satisfaction (RR = 1.8), cannabis use (RR = 1.8), being a victim of violence (RR = 2.1), and being a perpetrator of violence (RR = 2.8). Household dysfunction (compared to other single ACE category exposures) was most strongly associated with early sexual initiation (RR = 1.6), any mental health diagnosis (RR= 1.7), STI (RR = 2.2), and being incarcerated (RR = 2.3).

All possible two-way combinations of ACE categories (e.g., witnessing violence and child maltreatment; witnessing violence and household dysfunction; household dysfunction and child maltreatment) were significantly associated with all outcomes. The combined effect of household dysfunction and child maltreatment showed the strongest effect size, having the highest risk ratios in comparison to other two-way combinations (see Table 2a,b). The risk ratios for the co-occurrence of two ACE categories ranged from 1.2 for binge drinking in those exposed to both household dysfunction and child maltreatment to 5.0 for being a perpetrator of violence in those exposed to household dysfunction and child maltreatment.

Experiencing all three ACE categories before 18 years of age was associated with the highest levels of risk across all outcomes. Those who experienced all three ACE combinations were approximately two times more likely to smoke, binge drink, and to have been diagnosed with a mental illness as an adult; around three times more likely to have had early sexual intercourse and report lower life satisfaction; around four times more likely to report accidental teenage pregnancy and an STI diagnosis; almost six times more likely to engage in cannabis use, be a victim of violence, or have been incarcerated; and over seven times more likely to be a perpetrator of violence than those with no ACEs.

### 3.2. Additive Models

We assessed the additive and multiplicative risk of all possible two ACE category combinations. Specifically, we focused on how much the joint exposure of two ACE categories exceeded what would be expected when the associated effect of exposures on their own is summed. We also estimated the additive risk of all three ACE category combinations versus what would be expected when combining all three individual ACE category effects. The results revealed marked additive effects for violence-related outcomes (see Appendix A for all estimates). It is worth noting that not all ACE category combinations were additive. For example, there was no excess risk seen for life satisfaction as an outcome when participants had also experienced household dysfunction and witnessed violence. Other ACE category combinations saw less risk than was expected statistically (see Appendix A).

The proportion of people with no ACEs who were a victim of violence was 1.8% (see Figure 2a). This rose by 1.1% for those who witnessed violence, 1.4% for those exposed to household dysfunction, and 2.0% for those exposed to child maltreatment. In terms of two-way ACE category exposures, the most marked effect overall was seen for child maltreatment and household dysfunction. This combination saw a 5.4% rise in the proportion of people being a victim of violence, with 2.0% of this increase being due to the excess risk observed when these ACE categories co-occurred. The two ACE category combinations that conferred the most excess risk were household dysfunction and witnessing violence. The proportion of people who were a victim of violence increased by 4.9% above baseline for those who experienced household dysfunction and witnessed violence as children, with 2.4% of this increase being excess risk when these two ACE categories co-occurred.

The proportion of people who were a perpetrator of violence was 1.2% for those with no ACEs. This proportion rose by 0.1% for those who witnessed violence, 1.7% for those exposed to household dysfunction, and 2.2% for those exposed to child maltreatment (see Figure 2b). In terms of two-way ACE category exposures, the most marked effect overall was seen for child maltreatment and household dysfunction, showing a 4.8% rise above baseline, where 1.0% of this increase was due to excess risk when the two ACE categories co-occurred. The two ACE category combination that conferred the most excess risk was household dysfunction and witnessing violence. For those exposed to household dysfunction and witnessing violence as children, the overall proportion of people who were a perpetrator of violence rose by 4.3% above baseline, with 2.6% of this effect being due to excess risk when these ACE categories co-occurred.

Exposure to all three ACE categories conferred the most overall risk and excess risk. The proportion of those being a victim of violence rose by 8.4% above baseline for those exposed to all ACE categories (see Figure 2a), with 3.9% of this increase due to excess risk. The proportion of people being a perpetrator of violence rose by 7.7% above baseline, with 3.8% of this increase due to excess risk when all these ACE categories co-occurred (see Figure 2b).

## 4. Discussion

Data were analyzed from a large general population sample to investigate the effect of specific ACE categories in isolation and combination on experiences of violence, health-harming behaviors, and mental ill-health. To date, most ACE research has used cumulative ACE scores (e.g., 1, 2–3, 4+ ACEs) to estimate the increased risk associated with ACEs across health-related outcomes [7]. Using cumulative ACE scores precludes an ability to identify the most harmful ACEs individually or when combined. By exploring additive effects, we were able to estimate the proportion of people at risk across individual and multiple ACE category exposures. This highlighted that witnessing violence and household dysfunction was associated with a much higher prevalence of people being victims and/or perpetrators of violence than would be expected. This methodology allows us to pinpoint more pernicious ACE combinations that are difficult to identify from multiplicative models alone.

This research goes some way to overcome the equivalizing of ACEs, allowing identification of how much extra risk is present when different categories of ACEs co-occur.

Findings revealed that child maltreatment and household dysfunction in isolation exhibited strong associations across various outcomes. Specifically, child maltreatment, in comparison to other single ACE category exposures, was most strongly associated with accidental teenage pregnancy, low life satisfaction, cannabis use, being a victim of violence, and being a perpetrator of violence. Whereas, in comparison to other single ACE categories, household dysfunction was most associated with early sexual initiation, mental illness diagnosis, and incarceration.

Like previous findings, as the number of ACE category exposures increased, so did the associated risks for health-related outcomes. This effect was not equivalent across ACE category exposure combinations or outcomes. Additive modeling revealed the excess risks observed when ACE categories co-occurred. This was most marked for being a victim of violence and being a perpetrator of violence. Exposure to all three ACE categories was the most pernicious across outcomes.

Witnessing violence was associated with the least number of outcomes (victim of violence, being incarcerated, and cannabis use) and showed the weakest effect sizes in multiplicative models. However, witnessing violence conferred excess risk when experienced with other ACE categories. In simpler terms, when witnessing violence was combined with another ACE category, there was a higher proportion of people than expected being perpetrators and/or victims of violence. Witnessing domestic violence as a child has been associated with heterogenous outcomes [23]. Effective mother–child interactions and positive maternal mental health have been shown to promote resilience and positive outcomes. Thus, it might be the case that the addition of household dysfunction where parental mental health is affected in combination with witnessing violence reduces a child’s ability to be resilient. Witnessing violence and experiencing child maltreatment may provide the young person with a behavioral template of violence [24]. These experiences might make the young person more likely to be emotionally dysregulated [24]. This behavioral template and emotional dysregulation may amplify one’s likelihood of experiencing violence in later life. It is worth noting that those witnessing violence may be more likely to also experience more severe forms of child maltreatment and household dysfunction. As such, this could be a potential explanation as to why these additive effects are seen. The data analyzed here does not include information about the frequency of or the perceived severity of ACEs endured; thus, the extent to which this may contribute to the additive effects observed cannot be discerned. However, investigating the frequency and perceived severity of ACEs could be a promising direction for future research endeavors.

Efforts should be made to ensure children who have witnessed domestic violence are protected from additional ACE exposure. Household dysfunction was associated with poor outcomes such as early sexual initiation, mental illness diagnosis, and being incarcerated. Arguably being exposed to similar behaviors in the household, e.g., witnessing incarceration and growing up in an environment where a caregiver(s) experiences mental health issues is passed on to their children. However, it is worth noting that some of these outcomes could also be related to parental neglect, something that may co-occur alongside household dysfunction, but that was not measured here. Thus, we cannot estimate to what extent household dysfunction may reflect a wider pattern of neglect. It would be useful to identify the unique contribution of neglect to poor outcomes in isolation and in combination with other ACE categories. Child maltreatment alone was associated with a range of harms, especially those related to violence. Given its potential to affect a wide number of poor outcomes, targeted prevention should be prioritized, and appropriate support to help those exposed.

A recent state-of-the-art report about tackling ACEs suggests multiple ways of preventing and responding to ACEs [25]. Whilst several interventions focus on specific ACE types (e.g., domestic violence), overlapping themes exist amongst effective approaches, such as the implementation of laws that seek to protect young people from harm, supporting families, economic strengthening, education, and promoting a culture of respect and non-violent behavior [25]. Helping parents in terms of mental health, domestic violence, and substance use can also help children, preventing the transmission of future problematic health behaviors and outcomes. Effective intervention can mitigate future health-related costs that have been associated with ACEs [5,6,25]. Given the disproportionate level of harm associated with multiple ACE exposure observed in this study, interventions that mitigate these pernicious effects are likely to have significant economic and health benefits in the future. The finding that witnessing violence, when combined with another ACE category, amplifies the likelihood someone would become a victim of and/or perpetrator of violence, suggests that more work is needed to protect young people from violence in any form. The establishment of public health approaches to violence reduction (e.g., Wales Violence Prevention Unit) are practical approaches in the right direction [26]. Current efforts to make police services more trauma-informed are encouraged as there may be a disproportionate number of traumatized young people interacting with the criminal justice system [27]. We cannot estimate the severity or frequency of the violence perpetrated, but other work suggests committing acts of severe violence can also traumatize the individual [17]. As such, early violence prevention efforts may reduce the likelihood of children being retraumatized as adults, either via re-victimization or perpetration.

There are several limitations of this work. For example, the use of cross-sectional cohorts and retrospective questionnaires precluded any ability to make causal inferences. Retrospective questionnaires may illicit less accurate recall (being retrospective) [28]. The sample within this work represented those willing to participate or who were self-selecting, and as such, it may have missed harder-to-reach groups and those with minoritized identities. Furthermore, not all interviews were conducted in person; given that this study utilized data from a number of surveys, differing recruitment methods were adopted (see Appendix A), which may impact the validity of responses. Demographic variables such as sexual orientation and marital status were not included in all surveys and, thus, could not be accounted for here. Another potential limitation is the way ACEs have been categorized in this study. The witnessing violence ACE category included only one ACE exposure, whereas the child maltreatment and household dysfunction categories could include multiple. This limited our ability to interpret the true extent of witnessing violence versus other ACE categories, given that small effects could be due to only experiencing one ACE versus the type of ACE experienced. Future research may include other forms of witnessing violence, such as community violence. We also did not measure neglect across surveys, which is likely to also confer its own pattern of risk across outcomes.

Another aspect that limits the interpretation of the results is that we did not use the frequency of ACE exposure; thus, there was no way to disambiguate the effect of repeated exposure to ACE categories on the level of risk experienced across outcomes and whether this might also be affected by the age of exposure. Future work could work on exploring the impact of ACE category frequency (linear models) and investigate how frequency might confer more risk between and within ACE categories.

In recognizing the importance of diversity within research, it is crucial to acknowledge and address specific limitations associated with inclusivity in our research endeavors. There was limited ethnic variation in our sample, with representation being slightly lower (12%) than the national prevalence identified by the 2021 England and Wales census (19%). Furthermore, we were unable to explore differing effects across sexual and gender minorities due to data not being consistently collected across surveys. The effects of ACEs on these populations are developing areas of research [29], and future research should control for these effects in statistical models. There were some significant relationships between sociodemographic characteristics and outcomes which are provided in Appendix A. Analysis of demographic subgroups was beyond the scope of the current study, but understanding the prevalence of violence, health-harming behaviors, and mental ill-health across demographic groups would be an important area for future research.

Using ACE categories meant we could estimate the associated risk of ACEs on health-related outcomes and violence in isolation and when they co-occurred. Furthermore, by estimating additive relationships, we were able to identify the excess risk associated with ACE categories. Namely that witnessing violence was associated with the most excess risk over and above what would be expected when combined with another ACE.

## 5. Conclusions

Experiencing child maltreatment and household dysfunction in isolation was associated with increased risks across outcomes. Thus, exposure to either one of these ACE categories may confer a multitude of negative health-related outcomes. When multiple ACE categories co-occurred, there was evidence that their collective exposure surpassed expected statistical effects for violence-related outcomes. Given that witnessing violence amplifies risk when combined with other ACE categories, targeted interventions that protect young people from additional ACE exposures may mitigate a multitude of future harms. It is important to note that not all effects were additive and further work can be done to explore how the temporal ordering of ACE exposures may affect additive risks. The disproportionate risk associated with multiple ACE exposures suggests that effective interventions are needed to mitigate existing harm and reduce future issues for young people and the intergenerational transmission of violence.

## Figures and Tables

**Figure 1 ijerph-20-06633-f001:**
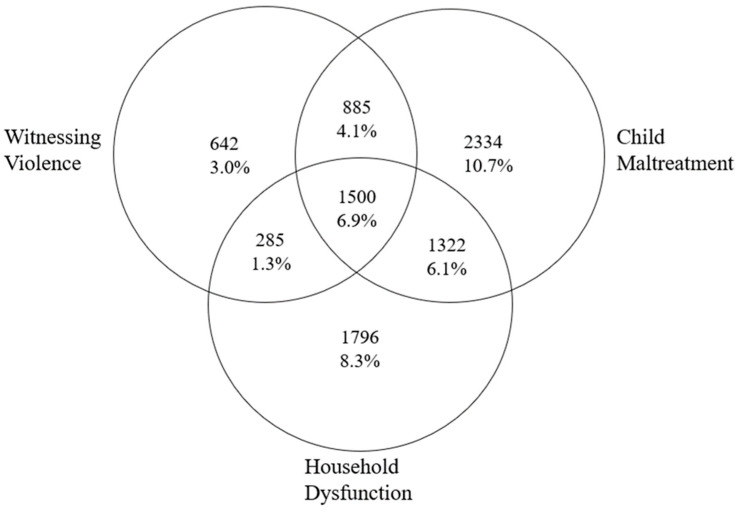
Venn diagram depicting the number of participants experiencing each ACE category. Total available sample (N = 21,716), No ACEs = 12,952 (59.6%).

**Figure 2 ijerph-20-06633-f002:**
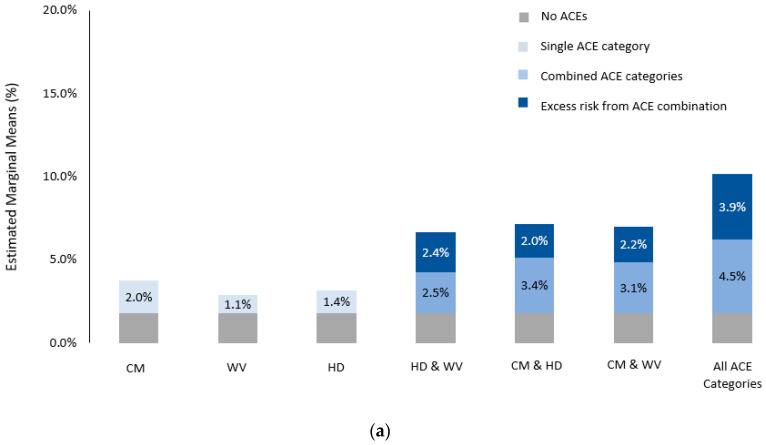
(**a**) Additive effects of ACE categories in isolation and combination—victim of violence; (**b**) additive effects of ACE categories in isolation and combination—perpetrator of violence. Note: CM = child maltreatment, WV = witnessed violence, HD = household dysfunction.

**Table 1 ijerph-20-06633-t001:** Demographic characteristics of the sample, distribution, and frequency.

Demography Grouping	*n*	%
All	21,716	100
Sex		
Male	9422	43.4
Female	12,294	56.6
Age		
18–29	4262	20.7
30–39	4067	19.7
40–49	4180	20.3
50–59	3959	19.2
60–69	4148	20.1
Ethnicity		
White	19,074	87.8
Other than White	2642	12.2
Deprivation quintile		
1—Least deprived	4596	21.2
2	4207	19.4
3	4278	19.7
4	4179	19.2
5—Most deprived	4456	20.5
Survey		
England 2012 (North West)	1421	6.5
England 2013	3885	17.9
England 2015 (South)	5454	25.1
Wales 2015	2028	9.3
Wales 2017	2497	11.5
England 2020–2021 (North West)	1819	8.4
Wales 2020–2021	2872	13.2

**Table 2 ijerph-20-06633-t002:** (**a**) Relative risk ratios for violence outcomes and substance use behaviors across ACE exposures (RR (95% CIs), *p* values). (**b**) Relative risk ratios for health-harming behaviors and mental health outcomes across ACE exposures (RR (95% CIs), *p*-values).

(a)						
ACE Exposure	Victim of Violence	Perpetrator of Violence	Incarcerated (Ever)	Smoking	Binge Drinking	Cannabis Use (Ever)
	*n* = 18,885	*n* = 18,876	*n* = 15,252	*n* = 21,689	*n* = 19,194	*n* = 21,631
Witnessed violence	1.6 [1.1, 2.5]	1.1 [0.6, 2.0]	2.0 [1.5, 2.8]	1.0 [0.9, 1.2]	1.2 [1.0, 1.5]	1.4 [1.2, 1.7]
	***p* = 0.029**	*p* = 0.857	***p* < 0.001**	*p* = 0.688	*p* = 0.112	***p* = 0.001**
Child maltreatment	2.1 [1.7, 2.6]	2.8 [2.2, 3.6]	2.1 [1.7, 2.6]	1.3 [1.2, 1.4]	1.2 [1.1, 1.4]	1.8 [1.6, 2.6]
	***p* < 0.001**	***p* < 0.001**	***p* < 0.001**	***p* < 0.001**	***p* < 0.001**	***p* < 0.001**
Household dysfunction	1.8 [1.4, 2.3]	2.4 [1.8, 3.1]	2.3 [1.9, 2.8]	1.2 [1.1, 1.4]	1.2 [1.1, 1.4]	1.7 [1.6, 1.9]
	***p* < 0.001**	***p* < 0.001**	***p* < 0.001**	***p* < 0.001**	***p* < 0.001**	***p* < 0.001**
Witnessing violence and child maltreatment	3.9 [3.1, 5.0]	4.3 [3.2, 5.6]	3.3 [2.7, 4.1]	1.5 [1.3, 1.7]	1.3 [1.1, 1.6]	1.8 [1.6, 2.1]
	***p* < 0.001**	***p* < 0.001**	***p* < 0.001**	***p* < 0.001**	***p* < 0.001**	***p* < 0.001**
Witnessing violence and household dysfunction	3.7 [2.6, 5.3]	4.6 [3.1, 6.8]	3.5 [2.4, 4.9]	1.5 [1.2, 1.8]	1.4 [1.1, 1.9]	2.0 [1.6, 2.6]
	***p* < 0.001**	***p* < 0.001**	***p* < 0.001**	***p* < 0.001**	***p* = 0.027**	***p* < 0.001**
Household dysfunction and child maltreatment	4.0 [3.3, 4.9]	5.0 [4.0, 6.2]	3.5 [2.9, 4.2]	1.6 [1.4, 1.8]	1.2 [1.1, 1.4]	2.6 [2.3, 2.8]
	***p* < 0.001**	***p* < 0.001**	***p* < 0.001**	***p* < 0.001**	***p* = 0.003**	***p* < 0.001**
All ACE categories	5.7 [4.9, 6.7]	7.4 [6.2, 8.9]	5.7 [4.9, 6.5]	2.0 [1.8, 2.2]	1.7 [1.5, 2.0]	2.9 [2.6, 3.2]
	***p* < 0.001**	***p* < 0.001**	***p* < 0.001**	***p* < 0.001**	***p* < 0.001**	***p* < 0.001**
(**b**)						
**ACE Exposure**	**STI (Ever)**	**Early Sexual Initiation**	**Accidental Teenage** **Pregnancy**	**Low Life Satisfaction**	**Any Mental** **Illness** **Diagnosis (Ever)**	
	***n* = 18,054**	***n* = 10,860**	***n* = 12,682**	***n* = 10,729**	***n* = 5293**	
Witnessed violence	1.2 [0.6, 2.4]	1.1 [0.8, 1.5]	1.4 [0.9, 2.0]	1.3 [1.0, 1.8]	1.2 [0.9, 1.6]	
	*p* = 0.615	*p* = 0.641	*p* = 0.105	*p* = 0.051	*p* = 0.202	
Child maltreatment	2.1 [1.5, 2.8]	1.7 [1.5, 1.9]	1.8 [1.4, 2.2]	1.8 [1.5, 2.1]	1.6 [1.4, 1.8]	
	***p* < 0.001**	***p* < 0.001**	***p* < 0.001**	***p* < 0.001**	***p* < 0.001**	
Household dysfunction	2.2 [1.6, 3.0]	1.6 [1.4, 1.9]	1.6 [1.3, 2.1]	1.6 [1.3, 2.0]	1.7 [1.5, 1.9]	
	***p* < 0.001**	***p* < 0.001**	***p* < 0.001**	***p* < 0.001**	***p* < 0.001**	
Witnessing violence and child maltreatment	3.4 [2.3, 4.9]	1.9 [1.5, 2.3]	2.8 [2.2, 3.7]	1.7 [1.3, 2.2]	1.6 [1.3, 1.9]	
	***p* < 0.001**	***p* < 0.001**	***p* < 0.001**	***p* < 0.001**	***p* < 0.001**	
Witnessing violence and household dysfunction	2.9 [1.6, 5.4]	1.9 [1.4, 2.6]	1.7 [1.0, 2.8]	2.0 [1.3, 2.9]	1.2 [0.8, 1.7]	
	***p* = 0.001**	***p* < 0.001**	***p* = 0.038**	***p* = 0.001**	*p* = 0.349	
Household dysfunction and child maltreatment	2.9 [2.1, 4.0]	2.2 [1.9, 2.5]	2.6 [2.1, 3.3]	3.0 [2.5, 3.5]	2.1 [1.9, 2.4]	
	***p* < 0.001**	***p* < 0.001**	***p* < 0.001**	***p* < 0.001**	***p* < 0.001**	
All ACE categories	4.2 [3.2, 5.5]	2.6 [2.3, 2.9]	3.9 [3.3, 4.6]	3.2 [2.7, 3.7]	2.1 [1.8, 2.3]	
	***p* < 0.001**	***p* < 0.001**	***p* < 0.001**	***p* < 0.001**	***p* < 0.001**	

Note: Significant effects in bold. ACE = adverse childhood experience.

## Data Availability

SPSS code will be made available upon request. Research data cannot be shared, given participants did not consent to share data beyond the research team.

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
