# Peer review of "ACEtimation—The Combined Effect of Adverse Childhood Experiences on Violence, Health-Harming Behaviors, and Mental Ill-Health: Findings across England and Wales"

_ijerph, 2023, doi:10.3390/ijerph20176633_

Round 1
Reviewer 1 Report
I thank the authors for the work they have done. Following the reading of your manuscript, I would like to suggest some ideas for its refinement:
- Please cite more literature associated with the outcomes of ACEs, such as body image disorders (Longobardi et al., 2022), depression and anxiety ( Racine et al, 2021), obesity (Wiss et al., 2020), etc... focus on recent meta-analyses.
- Pay more attention and care when describing the hypotheses.
- Perhaps more literature is needed (in particular, more recent literature) when discussing the effects of combining different ACEs and the individual ACEs considered.
- How did you ascertain the cognitive abilities of the participants? What requirements did they have to meet?
- were there any rewards for participating in the research?
- did you consider sexual orientation and marital status in the sociodemographic data? If not, why not?
- i find the association ACE-perpetrator of violence interesting. It should be better clarified that performing violent actions (even extreme ones) can be associated with ACE and in turn can constitute a traumatic aspect for the subject (Badenes-Ribera et al., 2021)
- limits, future research directions and practical implications should be further elaborated.
Suggested References
Longobardi, C., Badenes-Ribera, L., & Fabris, M. A. (2022). Adverse childhood experiences and body dysmorphic symptoms: A meta-analysis. Body image, 40, 267-284.
Racine, N., Devereaux, C., Cooke, J. E., Eirich, R., Zhu, J., & Madigan, S. (2021). Adverse childhood experiences and maternal anxiety and depression: a meta-analysis. BMC psychiatry, 21(1), 1-10.
Wiss, D. A., & Brewerton, T. D. (2020). Adverse childhood experiences and adult obesity: a systematic review of plausible mechanisms and meta-analysis of cross-sectional studies. Physiology & Behavior, 223, 112964.
Badenes‐Ribera, L., Molla‐Esparza, C., Longobardi, C., Sánchez‐Meca, J., & Fabris, M. A. (2021). Homicide as a source of posttraumatic stress?: A meta‐analysis of the prevalence of posttraumatic stress disorder after committing homicide. Journal of Traumatic Stress, 34(2), 345-356.
Author Response
Reviewer 1
- Please cite more literature associated with the outcomes of ACEs, such as body image disorders (Longobardi et al., 2022), depression and anxiety (Racine et al, 2021), obesity etc... focus on recent meta-analyses.
Dear reviewer, thank you for kindly providing these useful references. We agree with your comments and have added further information as per you recommendation.
See pages 1-2, lines 38-49
“ACEs have been associated with increased risk for engaging in health-harming behaviors such as alcohol consumption, drug use, sexual risk taking and interpersonal violence [2] . They have been found to be strongly associated with mental health conditions such as depression and anxiety [2], [3], and have also been linked to weight-related disorders such as dysmorphic disorder [4]and obesity [4]. Exposure to ACEs can subsequently impact physical health throughout the life course, including through early development of non-communicable disease such as cancer and cardio-vascular disease [2] Accordingly, evidence suggests greater use of health services in adults with ACEs [2], [5], [6]. There is also evidence indicating the intergenerational transmission of ACEs, along with maternal mental health issues post-partum [3]. Thus, ACEs not only affect the individuals who experience them but can potentially impact their own children.”
- Pay more attention and care when describing the hypotheses.
We apologize for these not being clearer in the first instance. As per the suggestion from the reviewer we have made more explicit what we expected to find and how this would be determined.
See page 3, lines 104-112
“Our primary hypothesis was that exposure to any individual ACE category or multiple ACE categories would be significantly associated with experiences of violence, health- harming behaviors and mental ill-health. We expected the strength of this relationship to vary depending on the combination of ACE categories experienced and the type of outcome being measured. We had no specific predictions as to which ACE categories would be most/least associated with any of the measured outcomes. The secondary hypothesis was that there would be an additive effect when ACE categories co-occurred; specifically, when ACE categories were experienced simultaneously the measured effects would be over and above those expected statistically.”
- Perhaps more literature is needed (in particular, more recent literature) when discussing the effects of combining different ACEs and the individual ACEs considered.
We thank the reviewer for this suggestion and have now incorporated more recent literature into the introduction:
See page 2, lines 67-70
“A similar study utilizing a large US national sample explored again how ACEs could amplify the negative effects of each ACE. Here, experiences of sexual abuse and physical abuse were associated most strongly with behavioral problems in young people [12].”
See page 2, lines 80-83
“For example, ACEs have been categorized into child maltreatment and household dysfunction, where the former includes more direct forms of harm or deprivation, and the latter includes disruptive environments and thus is an indirect form of ACE [14].”
See page 2, lines 83-87
“A recent study exploring the differential effects of ACE categorizations, namely child maltreatment and household dysfunction, found that the former was more predictive of depression and anxiety [14]. This highlights that different ACE categories may confer differing levels of risk which highlights the benefit of analyzing ACEs by category.”
- How did you ascertain the cognitive abilities of the participants? What requirements did they have to meet?
For face-to-face and telephone interviews, market research companies were commissioned to conduct collect data using professionally trained interviewers operating in adherence to the market research society code of conduct (we have now clarified this in the methods, see excerpt below). Interviewers are trained to identify whether potential participants were cognitively able to participate and used the exchange with potential participants at initial contact to assess this. They gave participants information about the study and sought informed consent prior to conducting the survey. If it became clear at any point the participant could not understand the information or had any difficulties understanding the content, they would not have proceeded with the interview. Online samples were self-completed, with study information provided prior to study initiation. Those cognitively unable to participate would be unlikely to provide data in full or proceed further than reviewing the information sheet.
Page 3 line 118-120
“Professional market research companies were commissioned to conduct household and telephone surveys, with interviewers operating to the Market Research Society Code of Conduct."
- Were there any rewards for participating in the research?
No incentive was provided for participants in the face to face or telephone surveys. However, online surveys recruited participants via online panels, which provide financial compensation for their time. We have now clarified this in the methods:
See page 3, lines 128-131
“Recruitment for online surveys used a commercial online panel, consisting of individuals who engage in online research for compensation. No incentives were provided for face-to-face or telephone interviews.”
- Did you consider sexual orientation and marital status in the sociodemographic data? If not, why not?
We agree with the reviewer that these are important variables to explore. However, these questions were not asked in all surveys precluding the ability to analyze such effects. We mention this as a potential limitation in the method section:
See page 12, 407-409
“Demographic variables such as sexual orientation and marital status were not included in all surveys, and thus could not be accounted for here.”
- I find the association ACE-perpetrator of violence interesting. It should be better clarified that performing violent actions (even extreme ones) can be associated with ACE and in turn can constitute a traumatic aspect for the subject (Badenes-Ribera et al., 2021)
This is an interesting point raised by the reviewer. We have referred to this in the discussion section:
See page 12, lines 395-399
“We cannot estimate the severity or frequency of the violence perpetrated but other work suggests committing acts of severe violence can also have traumatizing effect on the individual [28]. As such, early violence prevention efforts may reduce the likelihood of children being retraumatized as adults, either via re-victimization or perpetration.”
- limits, future research directions and practical implications should be further elaborated.
We have made the following additions to the discussion section:
See page 11, lines 351-362
“Witnessing violence and experiencing child maltreatment may provide the young person with a behavioral template of violence [24]. These experiences might make the young person more likely to be emotionally dysregulated [24]. This behavioral template and emotional dysregulation may amplify one’s likelihood of experiencing violence in later life. It is worth noting that those witnessing violence may be more likely to also experience more severe forms of child maltreatment and household dysfunction. As such, this could be a potential explanation as to why these additive effects are seen. The data analyzed here does not include information about the frequency of, or the perceived severity of ACEs endured. Thus, the extent of which this may contribute to the additive effects observed cannot be discerned. However, investigating the frequency and perceived severity of ACEs could be a promising direction for future research endeavors.”
See page 12, lines 388-399
“The finding that witnessing violence when combined with another ACE category amplified the likelihood someone would become a victim of and or perpetrator of violence, suggests more work is needed to protect young people from violence in any form. The establishment of public health approaches to violence reduction (e.g., Wales Violence Prevention unit) are practical approaches in the right direction [26]. Current efforts to make police services more trauma informed are encouraged as there may be a disproportionate number of traumatized young people interacting with the criminal justice system [27].”
Reviewer 2 Report
Dear Authors,
I have read the manuscript with interest, and I would like to provide constructive feedback that I hope will help to further improve the quality and impact of your manuscript:
- Data Collection Method: The authors should acknowledge as a limitation the fact that not all interviews were conducted in person. The potential impact of this on the data quality and the reliability of the findings should be discussed.
- Interpretation of Additive Effect: In the discussion section, it would be beneficial if the authors could provide a more detailed interpretation of the additive effect of witnessing violence on other factors. This could help readers better understand the implications of these findings.
- Categorization of ACEs: The decision to group insults, physical assaults and sexual harassment into a single category could be contested. It might be more informative to separate these two types of ACEs, as they could have different impacts on the outcomes under study.
- Potential Confounding Factors: The authors should consider the possibility that individuals who have experienced more severe forms of violence or household dysfunction may be more likely to witness violence than others. This could potentially confound the observed additive effect of witnessing violence.
- Ethnicity and Socio-economic Status: While the authors have controlled for ethnicity and deprivation quintile in their analyses, it might be beneficial to conduct separate analyses for different ethnic and socio-economic groups. This could provide a more nuanced understanding of the impact of ACEs within these specific populations.
Overall, the manuscript provides valuable insights into the impact of ACEs on various outcomes. However, addressing these points could strengthen the study and provide a more nuanced understanding of the findings.
Author Response
- Data Collection Method: The authors should acknowledge as a limitation the fact that not all interviews were conducted in person. The potential impact of this on the data quality and the reliability of the findings should be discussed.
We thank the reviewer for this observation and as per their recommendation we had added this as limitation to the discussion section.
See page 12, lines 404-409
“Furthermore, not all interviews were conducted in person, given that this study utilizes data from a number of surveys over time, differing recruitment methods were adopted (see Appendix A table A1) which may impact the validity of responses.”
2. Interpretation of Additive Effect: In the discussion section, it would be beneficial if the authors could provide a more detailed interpretation of the additive effect of witnessing violence on other factors. This could help readers better understand the implications of these findings.
This is an important point the reviewer raises, and we have made sure to address this within the discussion section of the manuscript. We have explained more generally in the discussion section the benefit of exploring additive effects as well as multiplicative ones.
See page 10, lines 319-325
“By exploring additive effects, we were able to estimate the proportion of people at risk across individual and multiple ACE category exposures. This highlighted that witnessing violence and household dysfunction were associated with much higher prevalence of people being victims and/or perpetrators of violence than would be expected. This methodology allows us to pinpoint more pernicious ACE combinations that are difficult to identify from multiplicative models alone.”
We have also provided a more detailed interpretation of the additive effect of witnessing violence on other factors.
See page 11, lines 351-355
“Witnessing violence and experiencing child maltreatment may provide the young person with a behavioral template of violence [24]. These experiences might make the young person more likely to be emotionally dysregulated [24]. This behavioral template and emotional dysregulation may amplify one’s likelihood of experiencing violence in later life.”
3. Categorization of ACEs: The decision to group insults, physical assaults and sexual harassment into a single category could be contested. It might be more informative to separate these two types of ACEs, as they could have different impacts on the outcomes under study.
We appreciate the authors comments here and agree that this could be contested. However, these variables are commonly categorized under the umbrella term of child maltreatment and are direct forms of abuse to the child. We have expanded upon this in the introduction why we have categorized it in this way.
See page 2, lines 78-87
“Researchers have categorized ACEs into subtypes to allow nuanced analyses whilst maintaining a pragmatic approach to data analysis and interpretation. For example, ACEs have been categorized into child maltreatment and household dysfunction, where the former includes more direct forms of harm or deprivation, and the latter includes disruptive environments and thus is an indirect form of ACE [14]. A recent study exploring the differential effects of ACE categorizations, namely child maltreatment and household dysfunction, found that the former was more predictive of depression and anxiety [14]. This highlights that different ACE categories may confer differing levels of risk which highlights the benefit of analyzing ACEs by category.”
See pages 2-3, lines 92-94
“There is likely to be cross over in these experiences, and previous studies have found subtypes of child maltreatment to be highly correlated with good convergent validity across measures [17].”
4. Potential Confounding Factors: The authors should consider the possibility that individuals who have experienced more severe forms of violence or household dysfunction may be more likely to witness violence than others. This could potentially confound the observed additive effect of witnessing violence.
We thank the reviewer for their important comment here. This is a very good point, and this is why we controlled for the impact of other categorizations in our analyses. For example, the way in which we designed our variables meant that each individual and simultaneous exposure controlled for the other effects not within that category. As we intended to identify the added risk associated with multiple exposure this could be a potential explanation and we have added the following to the discussion section:
See page 11, lines 355-359
“It is worth noting that those witnessing violence may be more likely to also experience more severe forms of child maltreatment and household dysfunction. As such, this could be a potential explanation as to why these additive effects are seen. The data analyzed here does not include information about the frequency of, or the perceived severity of ACEs endured. Thus, the extent of which this may contribute to the additive effects observed cannot be discerned. However, investigating the frequency and perceived severity of ACEs could be a promising direction for future research endeavors.”
5. Ethnicity and Socio-economic Status: While the authors have controlled for ethnicity and deprivation quintile in their analyses, it might be beneficial to conduct separate analyses for different ethnic and socio-economic groups. This could provide a more nuanced understanding of the impact of ACEs within these specific populations.
We thank the reviewer for this comment, and we agree that understanding how ACEs influence outcomes in different demographic groups is an important area for future research. We feel it is beyond the scope of this specific paper but have now raised it as an area for future studies. We have also now included the results for demographic variables from our multivariate models in an appendix table for information and referred to these in the text:
See page 5, lines 188-189
“(see Appendix D table A4a/b for analyses by demographic variables).”
See page 13, lines 430-435
“There were some significant relationships between sociodemographic characteristics and outcomes which are provided within Appendix D tables 4a/b. Analysis of demographic subgroups was beyond the scope of the current study but understanding prevalence of violence, health-harming behaviors & mental ill-health across demographic groups would be an important area for future research.”
6. Overall, the manuscript provides valuable insights into the impact of ACEs on various outcomes. However, addressing these points could strengthen the study and provide a more nuanced understanding of the findings.
We thank the reviewer for their kind comments, we also believe that after having addressed such comments the paper is much clearer than it was prior.
Round 2
Reviewer 1 Report
Thanks In my opinion, the manuscript can be accepted in the current form. The authors have successfully addressed all my concerns.